# Relationship Analysis of PM$_{2.5}$ and BLH using an Aerosol and Turbulence Detection Lidar

Chong Wang[1,2,*], Mingjiao Jia[2,*], Haiyun Xia[1,2], Yunbin Wu[1], Tianwen Wei[1], Xiang Shang[1], Chengyun Yang[1], Xianghui Xue[1], Xiankang Dou[1,3]

[1]CAS Key Laboratory of Geospace Environment, University of Science and Technology of China, Hefei, 230026, China
[2]Glory China Institute of Lidar Technology, Shanghai, 201315, China
[3]School of Electronic Information, Wuhan University, Wuhan, 430072, China
[*]These authors contributed equally to this work.

*Correspondence to*: Haiyun Xia (hsia@ustc.edu.cn)

**Abstract.** The atmospheric boundary layer height (BLH) is a key parameter in weather forecast and air quality prediction. To investigate the relationship between BLH and air pollution under different conditions, a compact micro-pulse lidar integrated both direct detection lidar (DDL) and coherent Doppler wind lidar (CDWL) is built. This hybrid lidar is operated at 1.5 μm which is eye-safe and is made of all-fiber components. The BLH can be determined from aerosol density and vertical wind independently. During a 45-hour continuous observation in June 2018, stable boundary layer, residual layer and convective boundary layer are identified. Fine structure of aerosol layers, drizzles and vertical wind near cloudbase are also detected. In comparison, the standard deviation between BLH values derived from DDL and CDWL is $0.06$ km, indicating the accuracy of this work. The retrieved convective BLH is a little higher than that from ERA5 reanalysis due to different retrieval methods. Correlation between different BLH and PM$_{2.5}$ is strongly negative before a precipitation event and become much weaker after the precipitation. Different relations between PM$_{2.5}$ and BLH may result from different BLH retrieval methods, pollutant sources and meteorological conditions.

## 1 Introduction

In recent decades, with rapid urbanization, air pollution has become a severe environmental problem in China (Chan and Yao, 2008; Li et al., 2016; Song et al., 2017; Zhang et al., 2012). Particulate matter (PM) with aerodynamic diameter less than 2.5 μm (PM$_{2.5}$), attracts public attentions due to its adverse effects on human health and environment (Brunekreef and Holgate, 2002; Cohen et al., 2017; Huang et al., 2014; Kampa and Castanas, 2008). In addition to pollutant emissions and topographic conditions, the spatial and temporal distribution of PM is mainly affected by meteorological conditions in troposphere, especially in atmospheric boundary layer (ABL) (Chen et al., 2018; Li et al., 2017c; Song et al., 2017; Su et al., 2018; Wei et al., 2018).

The ABL, also called PBL (planetary boundary layer), plays an important role in lower troposphere. ABL is directly influenced by and responds to the Earth's surface activities, such as frictional drag, evaporation, transpiration and heat

transfer, with a timescale of an hour or less (Stull, 1988). The ABL consists of three major parts during the diurnal evolution: convective boundary layer (CBL), stable boundary layer (SBL) and residual layer (RL) (Stull, 1988). The ABL is a key factor in control and management of air quality, numerical weather prediction, urban and agricultural meteorology, aeronautical meteorology, hydrology and so on (Large et al., 1994). Pollutants or any constituents within this layer is fully

mixed and vertically dispersed due to convection or mechanical turbulence (Seibert, 2000). The boundary layer height (BLH) determines the volume available for pollution dispersion and transport in the atmosphere. Low BLH and weak turbulence strengthen the accumulation of air pollutants (Miao et al., 2018; Petaja et al., 2016). Hence, the BLH is one of the fundamental parameters in dispersion models. A roughly anti-correlation relationship between $PM_{2.5}$ and BLH have been found in recent years (Du et al., 2013; Miao et al., 2018; Petaja et al., 2016; Su et al., 2018). However, the relationship

analysis of $PM_{2.5}$ and BLH in different ABL categories, i.e., aerosol derived (static) BLH and turbulence derived (dynamical) BLH, is still rare. Therefore, continuous observation of the BLH with high temporal and spatial resolution and the relationship between pollutions and BLH is desirable for air quality prediction.

There are always significant changes in vertical profiles of aerosol concentration, specific humidity, potential temperature or turbulence around the top layer of ABL, making it possible to derive the BLH. There are several instruments used for the

determination of BLH based on the sharp gradient in the vertical profiles mentioned above (Baars et al., 2008; Bonin et al., 2018; Li et al., 2017a; Seibert, 2000; Yang et al., 2017). For example, in-situ instruments, such as radiosonde, balloon, mast and aircraft, and remote sensing instruments, such as sodar, wind profiler, lidar and ceilometer. All of these instruments have advantages and shortcomings regarding accuracy, detection range, spatial and temporal resolution as summarized by Seibert (2000). Among these instruments, lidar system provides backscattering signal with sufficient spatial and temporal resolution,

long detection range and high accuracy to determine the BLH. These qualities make lidar a powerful tool for BLH assessment.

In recent decades, lidars are widely used in lower atmosphere via Mie scattering, Raman scattering and differential absorption (Campbell et al., 2002; Godin et al., 1989; Murray and van der Laan, 1978; Reitebuch, 2012; Renaut et al., 1980; Xia et al., 2007). Aerosol, trace gas concentration, atmospheric density, temperature and wind can be detected by these lidars.

Recently, a micro-pulse direct detection lidar (DDL) based on up-conversion technology was developed to make continuous measurements of aerosol in troposphere (Xia et al., 2015). A coherent detection Doppler wind lidar (CDWL) was developed to measure wind field in ABL (Wang et al., 2017). Different from traditional micro-pulse lidars operated at or near 532 nm (He et al., 2008; Li et al., 2017b; Sawyer and Li, 2013), these two lidars are operated at 1.5 μm, which are eye-safe and can be made with all-fiber components. The 1.5 μm laser shows the highest maximum permissible exposure in the wavelength

range from 0.3 to 10 μm (Xia et al., 2015). The invisible infrared eye-safe laser makes these two lidars can work in a densely populated city horizontally. The all-fiber structure makes these lidars robust, immune to external environment changes such as vibration and temperature. Based on these two lidars, BLH values can be derived from both aerosol density and turbulence. The simultaneous implementation of DDL and CDWL will improve the precision of BLH assessment and enrich the meteorological data in ABL.

Generally, DDL and CDWL belong to different lidar categories. In this work, a hybrid lidar integrating both systems are developed for simultaneous measurements of aerosol and vertical wind. The integrated lidar is utilized to further understand the relationship between PM$_{2.5}$ and BLH. The integrated lidar system, meteorology and PM data are described in Sect. 2. The retrieval methods of BLH are briefly introduced in Sect. 3. In Sect. 4, the results and discussions on lidar data, the retrieved

BLH and the relationship between PM$_{2.5}$ and BLH are presented. Finally, the conclusion and summary are drawn in Sect. 5.

## 2 Instruments and Data

### 2.1 The integrated lidar system

A compact and integrated micro-pulse lidar system is developed. Benefiting from all-fiber configuration and up-conversion technology, the lidar inherits advantages from both DDL and CDWL. The DDL is based on up-conversion technology and is

used for long-range aerosol measurement (Xia et al., 2015). The CDWL is used for wind field measurement (Wang et al., 2017). Two lidar systems use only one set of laser source, optical collimator and control system. The unique optical telescope guarantees that the measured signal in both systems are from the same backscattering volume, and the radial wind profile and aerosol concentration are measured simultaneously.

The diagram of the system is shown in Fig. 1. The seed laser emits continues wave (CW) at 1548 nm, then the CW is split

into local oscillator and transmitted seed laser by a beam splitter (BS). The transmitted seed laser is chopped, and frequency shifted 80 MHz by an acoustic-optical modulator (AOM). After the AOM, the transmitted laser is amplified by an Erbium doped fiber amplifier (EDFA) and transmitted into the atmosphere via a collimator. The pulse duration is 300 ns and the pulse energy is 110 μJ. The atmospheric backscattering is collected by two telescopes, adopting double 'D' configuration. As shown in Fig. 1, the two aspheric lenses are glued together with parallel optical axes for easy alignment and avoiding blind

zone. The absolute overlap distance is 1 km. On the CDWL channel, the backscattering pass through a circulator is chopped by an optical switch (OS), which cuts off the reflection light from the circulator and lens. The reflection light is much higher than the atmospheric backscattering and will cause saturation and even breakdown in the balanced detector (BD). After the OS, the backscattering is mixed with local oscillator and measured on the BD. The analog signal is converted to digital signal by an ADC and then processed by a PC. On the DDL channel, the backscattering is collected and mixed with a pump

laser at 1950 nm in a wavelength division multiplexer (WDM). The mixed laser then passes through the periodically poled lithium niobate waveguide (PPLN). The backscattering at 1548 nm is converted to 863 nm by the PPLN and detected by a Silicon single-photon detector (SPD). A filter is used to filter out the noise. A multi-channel scaler (MCS) records the digital signal. Benefiting from coherent detection and the narrow passband of the PPLN, this integrated lidar can perform all-day detection of the atmosphere. The detailed parameters of the integrated lidar are listed in the Table 1.

During the experiment, the integrated lidar is pointed vertically. Then the vertical wind and backscattering intensity are measured simultaneously. The raw DDL data is recorded with a spatial and temporal interval of 45 m and 2 s, respectively,

while CDWL data is 60 m and 2 s, respectively. The integrated lidar is deployed 5 m above ground on the campus of University of Science and Technology of China (USTC, 117.26 °E, 31.84 °N), an urban area in Hefei, China.

## 2.2 Meteorology and PM$_{2.5}$ data

A weather transmitter (Vaisala WXT520) is used to measure meteorological parameters, including temperature, relative humidity, liquid precipitation, barometric pressure, wind velocity and direction. A visibility sensor (Vaisala PWD50) is used to measure the atmospheric visibility. A wide range aerosol spectrometer (Grimm Mini WRAS 1371) measures aerosol volume size distribution ranging from 10 nm to 35 μm over 41 channels (Shang et al., 2018). Then PM$_{2.5}$ and PM$_{10}$ values are calculated. All these instruments were deployed 60 m above ground and 250 m east of the integrated lidar on the top of a research building in USTC. During the experiment, all these meteorological data are recorded with an interval of 1min.

## 3 BLH retrieval methods

The BLH is retrieved from both aerosol concentration and vertical wind in this experiment. For the DDL data, the range corrected lidar signal (RCS), $N(R)R^2$, has a sharp decrease at BLH, where $N(R)$ is the backscattering photon number from altitude of R. As for the CDWL, the temporal vertical velocity variation in the ABL is much stronger than that in the free atmosphere. The carrier to noise ratio (CNR) of the CDWL also represents the aerosol concentration, which can also be used to determine the BLH.

Haar wavelet covariance transform (HWCT) method is used to retrieve BLH from aerosol concentration. The HWCT $W_f(a, b)$ is defined as (Brooks, 2003):

$$W_f(a, b) = \frac{1}{a} \int_{z_b}^{z_t} f(z) \, h\left(\frac{z - b}{a}\right) dz \quad , \tag{1}$$

with the Haar function:

$$h\left(\frac{z - b}{a}\right) = \begin{cases} 1, b - \dfrac{a}{2} \le z \le b \\ -1, b \le z \le b + \dfrac{a}{2} \\ 0, elsewhere \end{cases} . \tag{2}$$

Where $f(z)$ is the normalized RCS or CNR, $z_b$ and $z_t$ are the bottom and top height of a selected range, a is the dilation of the Haar wavelet and b is the center position of the Haar function. For a given dilation, the height where maximum $W_f(a, b)$ appears is considered to be the BLH. Considering different vertical spatial resolutions and having tested multi values of dilation, a dilation of 150 m and 250 m is applied for RCS and CNR, respectively for this 45-hour observations. Compared with gradient method, HWCT method has greater adjustability and robustness (Korhonen et al., 2014). The interference by multi aerosol layers in the ABL is negligible for an appropriate dilation. In order to reduce the interference from unexpected

noise, the signal is averaged to a temporal resolution of 1 min in BLH determination. It should be noted that cloud layer could affect the BLH results. A top-limit is set to the HWCT method for higher clouds. For the scattered stratocumulus that may exist in the capping layer, the differences between cloud top and BLH are relatively small. In addition, the duration time of stratocumulus is also short in the field of view of the lidar. Thus the influence of scattered stratocumulus is negligible. The

low level cloud in the ABL can be identified by the paired minimum $W_f(a, b)$ and maximum $W_f(a, b)$ occurs at heights close to each other. The BLH cannot be retrieved under this condition. As described in Sect. 2.1, RCS should be corrected with an overlap factor before the analysis. As an example, the measured RCS and CNR after one-minute average (after overlap correction and background noise deduction) at 1 June 2018, 10:40 am is shown in Fig. 2a. The corresponding HWCT results are shown in Fig. 2b, from which the BLH can be determined.

The BLH can also be determined from the variance of vertical velocity $\sigma_w^2$, which represents the vertical component of the turbulence kinetic energy. For a given time window and a reliable threshold, below the BLH, the $\sigma_w^2$ is larger than the threshold, and vice versa. The threshold varies from different locations (Huang et al., 2016). In this study, the threshold is set to be 0.06 m$^2$s$^{-2}$ which is suitable as shown in Fig. 2c. A median algorithm is used to mitigate the interference and fluctuation from unexpected turbulence and noise in the free atmosphere. Step (1), select all the height with $\sigma_w^2$ less than the threshold;

Step (2), find the median height $z_m$ selected in Step (1); Step (3), the BLH is the maximum height below $z_m$ that $\sigma_w^2$ larger than the threshold. An example of employing this threshold and algorithm is shown in Fig. 2c. Some confusing points, such as that at ~1.0 km and ~1.6 km in Fig. 2c can be distinguished.

Reanalysis data is always used in climatological and regional analysis of BLH (Collaud Coen et al., 2014; Guo et al., 2016; Seidel et al., 2012). ERA5 is the newest generation of ECMWF (European Centre for Medium-Range Weather Forecasts)

atmospheric reanalysis of the global climate. ERA5 reanalysis assimilates a variety of observations and models in 4-dimensional. The data has 137 levels from the surface up to 80 km altitude, the horizontal resolution is 0.3° for both longitude and latitude (Hersbach and Dee, 2016). The hourly BLH from high resolution realisation sub-daily deterministic forecasts of ERA5 is used to cross-check the BLH retrieved from lidar since there is no sounding data in Hefei. The BLH in ERA5 is determined by the bulk Richardson number ($Ri_b$) method (ECMWF, 2017; Seidel et al., 2012; Vogelezang and

Holtslag, 1996). The bulk Richardson number $Ri_b$ is defined as (Vogelezang and Holtslag, 1996):

$$Ri_b = \frac{gh(\theta_{vh} - \theta_{v0})}{\theta_{v0}(u_h{}^2 + v_h{}^2)} \qquad . \tag{3}$$

Here $g$ is the acceleration of gravity, $h$ is height, $\theta_{v0}$ and $\theta_{vh}$ are the virtual potential temperature at the surface and $h$, $u_h$ and $v_h$ are component wind speeds at h, respectively. The BLH is then defined as the lowest height where the $Ri_b$ reaches a critical value of 0.25 (ECMWF, 2017).

Compared to the BLH retrieval, RL top can be identified through a simply rough threshold which is described in the

Appendix.

## 4 Results and discussion

### 4.1 Observational results

Figure 3 shows a continuous observation over 45 hours from 1 June to 2 June in 2018. The RCS with 1 min temporal resolution, CNR and vertical wind with 20 s temporal resolution are shown, respectively. The black dotted line in each panel indicate the BLH derived from RCS, CNR and vertical wind, called as $BLH_{RCS}$, $BLH_{CNR}$ and $BLH_{VAR}$ in this study. Meanwhile, the red dotted lines in each panel indicate the RL tops. Sunrise and sunset times at local time (LT) of 05:06 and 19:12 are marked by red triangles and blue inverted triangles. The experimental observation ends with rainfall on the ground at ~2 June, 21:00 LT.

As shown in Fig. 3a, aerosol layer experiences a significant diurnal cycle within a height of 2 km. Before 1 June, 09:00 LT, an aerosol derived SBL caused by radiative cooling from the ground can be easily found below ~0.7 km with higher aerosol concentration than that in the RL. Subsequently, the ABL starts to grow due to solar heating after sunrise and deepens to a maximum height of about 2km in mid-afternoon. Sporadic stratocumulus appears at the top of the ABL with strong backscattering signal. During the night from 1 June, 22:00 to 2 June, 06:00, the backscattering increases in the RL, which is related to the increase of aerosol concentration. After the sunrise on June 2, the ABL grows as it on June 1, but the BLH is lower than that on June 1.

The CNR measured by CDWL is shown in Fig. 3b. The evolution of ABL is similar to that of RCS. The phenomena that observed in RCS described above can be also found in CNR. Figure 3c shows the height-time cross section of vertical wind. To guarantee the precision of the wind measurements, the data with CNR below -35 dB is abandoned (Wang et al., 2017). The downward vertical wind is positive and vice versa. Obviously, the convective ABL is well mixed with strong turbulence during the daytime between sunrise and sunset. Wave-like motions also exist in the nocturnal ABL associated with stratified atmosphere.

Cloud with strong backscattering can be detected between ~3 and ~9 km height by both DDL and CDWL. Corresponding vertical velocity of cloud is measured by the CDWL. Fine cloud structures above the ABL are shown in RCS with high spatial and temporal resolution due to higher detection efficiency. In addition, in the height ranging from ~3 to ~6 km on June 1, several transport aerosol layers can be detected in RCS despite accompanying sunshine induced noise. Interestingly, the transport aerosol layers meet the cloudbase during the night on June 1. The fine structures around cloudbase suggest the existence of drizzles. Moreover, precipitation in cloud can be identified by assuming that precipitation has a fall velocity greater than 1 m s$^{-1}$ (Manninen et al., 2018). A precipitation case is indicated by the red arrow in Fig. 3c at approximately 2 June, 02:00 LT. The $PM_{2.5}$ value measured simultaneously is illustrated as the brown dotted line in Fig. 3c. A sharp increase of $PM_{2.5}$ occurs during the precipitation. These results hint the potential applications of this integrated lidar in the investigations of aerosol-cloud-precipitation interactions.

The simultaneous measurements of meteorological parameters including temperature, pressure, wind velocity, wind direction, visibility and relative humidity near the ground are shown in Fig. 4a to c. It should be noted that the building

where the instrument deployed would have an impact on these meteorological parameters. There is no precipitation event recorded on the ground by weather transmitter, even during the precipitation in the cloud as shown in Fig. 3c. Weak wind condition (velocity less than 4 m s$^{-1}$) during the whole experiment is not conducive to the aerosol diffusion near the ground. The wind direction is always northly despite the easterly wind before sunrise on June 1. Two haze event occurred, with visibility less than 10 km and relative humidity less than 80%, during this experiment (Administration, 2010). As mentioned before, there is a sudden increase in PM$_{2.5}$ at approximately 2 June, 02:00 LT during the precipitation. There are also sudden changes in relative humidity and visibility at the same time indicated by the vertical dash-dotted lines. Simultaneous measurements of aerosol size volume distribution during this experiment is shown in Fig. 4d. The amount of aerosol rises in all size channels at ~2 June, 02:00 LT. Concentrations of PM$_{2.5}$ and PM$_{10}$ have almost the same evolution processes. It reveals that there is no specified single anthropogenic emission. The wet growth of the existing small particles caused by the precipitation above the ground may be responsible for the sudden increase of aerosols. Therefore, the experiment is chopped into two sections by the precipitation event, as the vertical dash-dotted lines in Fig. 4.

## 4.2 BLH retrieval results

As shown in Fig. 3, the BLH results are well retrieved, indicating that the HWCT and variance methods are appropriate for BLH determination. A top-limit of 2.5 km of BLH is applied during the BLH retrieval. A comparison is performed as shown in Fig. 5a with retrieved BLH$_{RCS}$, BLH$_{CNR}$, BLH$_{VAR}$ and BLH from ERA5 (BLH$_{ERA5}$). The BLH$_{RCS}$ and BLH$_{CNR}$ are smoothed with median value by a 5 min temporal window while the BLH$_{VAR}$ is smoothed by a 20 min temporal window. In general, there is a significant diurnal variation in BLH as expected. All three retrieved BLH from lidar measurements are comparable when the ABL is fully mixed. While in nocturnal ABL, the aerosol derived BLH$_{RCS}$ and BLH$_{CNR}$ are much higher than turbulence derived BLH$_{VAR}$ showing the different categories of SBL. A criterion is proposed to classify the ABL as CBL and RL/SBL by the values of BLH$_{VAR}$ and BLH$_{RCS}$ in this study. A parameter is defined as $\Delta = $ BLH$_{RCS} - $ BLH$_{VAR}$. The sign of $\Delta$ is positive at nighttime in most cases. In the evening, a SBL is capped by a RL as shown in Fig. 5a. In the morning, when BLH$_{VAR}$ meets the value of BLH$_{RCS}$, i.e., the sign of $\Delta$ become negative or the value of $\Delta$ is less than a specified value for the first time after midnight, the type of ABL changes from RL/SBL into CBL. In the Afternoon, when BLH$_{VAR}$ departs from BLH$_{RCS}$, i.e., the sign of $\Delta$ become positive or the value of $\Delta$ is greater than a specified value for the last time before midnight, the ABL turns into RL/SBL again. This diurnal evolution of ABL is similar to that described in Stull (1988) and Collaud Coen et al. (2014). During the RL/SBL, two kinds of SBL top (RL bottom) are classified by the BLH$_{RCS}$ and BLH$_{VAR}$. For the CBL, the BLH from lidar is a little higher than BLH$_{ERA5}$, especially during afternoon. This is in agreement with earlier analysis that climatological BLH based on Richardson's method is substantially lower than BLH derived from other methods (Seidel et al., 2010). While for the turbulence derived SBL, the BLH$_{VAR}$ is comparable with BLH$_{ERA5}$.

For a quantitative analysis, statistical comparisons of BLH$_{RCS}$ with BLH$_{VAR}$ and BLH$_{CNR}$ are visualized in Fig. 5b and c. The BLH$_{VAR}$ and BLH$_{CNR}$ are plotted versus the corresponding BLH$_{RCS}$. Scatter diagrams of data points almost lie on the blue

and red dashed lines that represent $BLH_{VAR} = BLH_{RCS}$ and $BLH_{CNR} = BLH_{RCS}$, respectively. Note that the $BLH_{RCS}$ is interpolated to the same time series of $BLH_{VAR}$ in Fig. 5b and only BLH in CBL is plotted. The $BLH_{CNR}$ agrees well with $BLH_{RCS}$, despite a difference in RL due to the elevated aerosol layer in the early morning on June 2. The differences between the two results show a standard deviation of 0.06 km of the Gauss fitting. For the CBL, the $BLH_{VAR}$ also agrees well with

$BLH_{RCS}$, with a standard deviation of 0.17 km through Gauss fitting.

## 4.3 Relationship between PM$_{2.5}$ and BLH

Recently, Su et al. (2018) and Miao et al. (2018) investigated the relationships between the BLH and surface pollutants in China. The influences of topography, seasonal variation, emissions and meteorological conditions on the BLH-PM$_{2.5}$ relationships were discussed. Nevertheless, due to the relatively low temporal resolution from space-borne lidar and

radiosonde measurements, the influence of different ABL types on the BLH-PM$_{2.5}$ relationships are rarely studied.

Figure 5d and e show the relationships between PM$_{2.5}$ concentration and BLH. The correlation coefficients between BLH ($BLH_{RCS}$ in CBL and RL/SBL, and $BLH_{VAR}$ in CBL) and PM$_{2.5}$ before and after precipitation in Fig. 5d and e are listed in Table 2. An obvious anti-correlation is shown before precipitation between BLH and PM$_{2.5}$ concentrations in both CBL and RL/SBL with correlation coefficient of ~ -0.9. An inverse fitting formula $PM_{2.5}=A+B/BLH$ is used to describe the PM$_{2.5}$-

BLH relationships in Fig. 5d. The resulted parameters of $A$, $B$ are listed in Table 2. The nonlinear inverse function shows good performance with coefficient of determination $R^2$=0.84, 0.65 and 0.85. In general, these results show good responses of PM$_{2.5}$ to aerosol derived BLH ($BLH_{RCS}$) evolution with larger $R^2$ and stronger correlation than turbulence derived BLH ($BLH_{VAR}$) both before and after precipitation. In addition, as shown in Fig. 5d, the inverse function of PM$_{2.5}$ to the $BLH_{RCS}$ and $BLH_{VAR}$ show good consistency in CBL. While in RL/SBL, the inverse function of PM$_{2.5}$ to the $BLH_{RCS}$ is quite

different from that in CBL. The parameter $A$ in the inverse fitting formula of the PM$_{2.5}$-BLH relationship for $BLH_{RCS}$ in RL/SBL is triple/twice as large as that for $BLH_{RCS}$/$BLH_{VAR}$ in CBL as listed in Table 2, while the parameter $B$ has similar values. This difference of parameter $A$ represents a higher PM$_{2.5}$ concentration in RL/SBL. After precipitation, as shown in Fig. 5e and Table 2, there are relatively weak anti-correlations of -0.34, -0.21 and -0.22, respectively. The relationships between BLH and PM2.5 are changed after precipitation. Recently, Geiß et al. (2017) investigated correlations between BLH

and concentrations of pollutants (PM$_{10}$, O$_3$, NO$_x$). They found that the correlations of BLH with PM$_{10}$ were quite different for different sites without showing a clear pattern. In addition, the reflection and absorption of the incoming solar radiation by the clouds on 2 June 2018 could also affect the diffusion of aerosols. Therefore, BLH with different retrieval methods, pollutant sources and meteorological conditions should be considered in air quality prediction models.

## 4.4 Aerosol-Cloud-ABL interaction

Moreover, the ABL in cloudy condition on June 2 grows slower with lower BLH than that in fair weather on June 1 as shown in Fig. 5a. The maximum BLH on June 2 is about 1.7 km high while the maximum BLH on June 1 is about 2.3 km high. In addition, the RL tops becomes lower when the cloud layer occurs around 4 km altitude on June 2. These phenomena

may be in relation to the aerosol-cloud-ABL interaction. There are several transport aerosol layers above ABL as shown in Fig. 3a. These transport aerosol layers may act as the cloud condensation nuclei during the cloud formation between ~3 and ~5 km on June 2. The clouds play important roles in earth's energy budget. As more incoming solar radiation is reflected and absorbed by clouds, less energy enters the ABL, resulted in weaker CBL development and lower BLH on June 2 than that on June 1. In addition, the weaker convection may lead to the higher aerosol concentration in ABL on June 2 as shown in Fig. 3a and 3(b). These results hinted a strong aerosol-cloud-ABL interaction during the ABL evolution.

## 5 Conclusion

A compact integrated lidar system that integrates both DDL and CDWL is demonstrated. The DDL incorporated a fiber laser at 1.5 μm and an up-conversion detector. This design of lidar makes it more eye-safe than traditional laser of 355, 532 and 1064 nm. All-fiber configuration is realized to guarantee the high optical coupling efficiency and robust stability. Two lidar systems use only one set of laser source, optical collimator and control system. Thus this integrated lidar can make simultaneous measurements of aerosol density, vertical wind and clouds with high spatial and temporal resolution.

The BLH values derived from aerosol and turbulence are determined from 45-hour continuous measurements. Two methods of HWCT and variance are employed in BLH determination, respectively. The BLH retrieved from different methods are comparable to each other, and so does the RL tops. The standard deviation between aerosol derived BLH from DDL and CDWL is 0.06 km. The BLH derived from vertical wind is comparable with BLH from ERA5 reanalysis data, also with larger BLH than ERA5 due to different retrieval methods as in other studies (Seidel et al., 2010). During the evolution of ABL, the clouds suppress the growth of ABL, leading aerosol increase in ABL. The variations of $PM_{2.5}$ and BLH before and after a precipitation event in clouds are analyzed in different ABL categories adopting different methods. There is a strong inverse relation between BLH and $PM_{2.5}$ in both CBL and RL/SBL before a precipitation. However, the relationship is relative weak after the precipitation. In addition, a good response of $PM_{2.5}$ to aerosol derived BLH evolution with larger $R^2$ and stronger correlation than turbulence derived BLH both before and after the precipitation.

The reasons for the differences in the relationships between BLH and $PM_{2.5}$ may result from both cloud effect and pollutant sources not just the precipitation. This required more data based on different instruments, such as horizontal wind field (Shangguan et al., 2017), temperature profiles (Mattis et al., 2002), and depolarization ratio of aerosol (Qiu et al., 2017) and pollutant components in ABL. To probe the mechanism of the BLH-$PM_{2.5}$ relations under different conditions, such as before and after the precipitation, not only such observations, but also model simulation are needed in further studies. The application of such an integrated lidar in this research will contribute to our understanding of ABL and aerosol-cloud-precipitation interactions. Thus improve our ability in weather forecast and air quality prediction in future.

*Data availability*

The ERA5 data sets are publicly available from ECMWF website at https://www.ecmwf.int/en/forecasts/datasets/reanalysis-datasets/era5, last access, 20 May 2019. Lidar and meteorological data can be downloaded from http://www.lidar.cn/datashare/Wang_et_al_2018c.rar, last access, 20 May 2019.

**Appendix: The RL top retrieval method**

Besides BLH, RL top is also important in model validation and parameterization development. A simple method to retrieve RL top from RCS, CNR and variance of vertical velocity profiles is proposed. In order to reduce the interference from noise, the RL top is determined with a temporal resolution of 5 min. Dominant aerosol layer tops are easy to be identified around 2 km altitude as shown in Fig, 3. Thus the aerosol layer tops are limited between 1 and 2.5 km altitude range. A threshold

method is suitable for RCS and CNR profiles. For this observation, the threshold is set to be $5 \times 10^{10}$ for RCS profile ($1 \times 10^{10}$ for resolution of 1 min as shown in Fig. 3a) and -30 dB for CNR profile. For profiles of variance of vertical velocity, the aerosol layer is identified as the altitudes under the minimum altitude where invalid data exists, e.g., ~1.6 km in Fig. 2c. If the difference between aerosol layer top and BLH is larger than a threshold, e.g., 0.3 km in current study, the aerosol layer top is identified as RL top. It should be noted that all the values of threshold used here may varies at different places for

different lidars. These values may be only suitable for during this observation.

*Author contribution*

HX and XD conceived and designed the study. CW, YW, TW performed the lidar experiments observations. XS performed the meteorological and PM observations. CY downloaded and analyzed ERA5 data. MJ and CW carried out the data analysis and prepared the figures, with comments from other co-authors. CW, MJ, HX and XX interpreted the data. CW, MJ and HX

wrote the manuscript. All authors contributed to discussion and interpretation.

*Competing interests.*

The authors declare that they have no conflict of interest.

*Acknowledgements.*

We acknowledge the use of ERA5 data sets from ECMWF website at https://www.ecmwf.int/en/forecasts/datasets/

reanalysis-datasets/era5. We like to thank Dr. Matthias Wiegner for his helpful suggestions. We are grateful to Shengfu Lin, Dr. Mingjia Shangguan and Jiawei Qiu for valuable discussions.

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

**Table 1. Key Parameters of the Integrated Lidar.**

| Parameter | CDL | DDL |
|---|---|---|
| Wavelength | 1548 nm | |
| Pulse Duration | 300 ns | |
| Pulse Energy | 110 μJ | |
| Repetition frequency | 10 kHz | |
| Diameter of collimator | 80 mm | |
| Diameter of telescope | 80 mm | 70 mm |
| Spatial resolution | 45 m | |
| Temporal resolution | 2 s | |
| Maximum range | 15 km | |
| Azimuth scanning range | 0 - 360 ° | |
| Zenith scanning range | 0 - 90 ° | |

**Table 2. The correlations between BLH and PM$_{2.5}$ and the inverse fitting results.**

| Time | Before precipitation | | | After precipitation | | |
|---|---|---|---|---|---|---|
| ABL | CBL | CBL | RL/SBL | CBL | CBL | RL/SBL |
| BLH | BLH$_{RCS}$ | BLH$_{VAR}$ | BLH$_{RCS}$ | BLH$_{RCS}$ | BLH$_{VAR}$ | BLH$_{RCS}$ |
| C[a] | -0.92 | -0.89 | -0.93 | -0.34 | -0.21 | -0.22 |
| A[b] | 7.98 | 12.46 | 23.16 | | | |
| B[b] | 42.22 | 35.92 | 32.65 | | | |
| R$^{2}$ [c] | 0.84 | 0.65 | 0.85 | | | |

[a] C: Correlation coefficients.

[b] A/B: The parameters of the inverse fitting formula $PM_{2.5}=A+B/BLH$.

[c] R$^2$: Coefficient of determination of the inverse fitting.

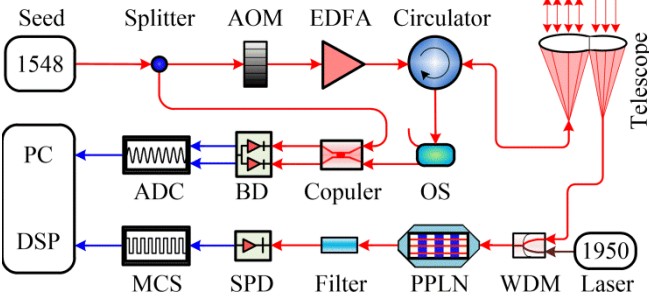

**Figure 1: The diagram of the integrated lidar system.**

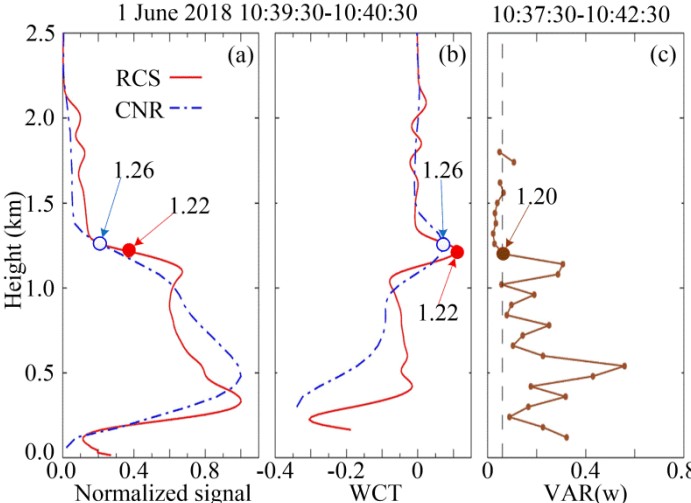

**Figure 2: (a) 1 min mean normalized RCS and CNR profiles. (b) The corresponding HWCT results of the RCS and CNR profiles in (a). (c) The vertical velocity variance profile. The black dashed line indicated the threshold of 0.06 m²s⁻². The red solid circle, blue circle and brown solid circle denoted by the arrows indicate the retrieved BLH, with the values at the end of the arrows.**

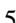

**Figure 3: Lidar observational results from 1 June 2018 to 2 June 2018. (a) The 1 min mean time series of logarithmic RCS profiles measured by DDL. The height time cross section of (b) CNR and (c) vertical wind measured by CDWL with 20 s temporal interval. The downward (upward) vertical wind is positive (negative). The brown dotted line indicates the PM$_{2.5}$ concentration near the ground. The black (red) dotted lines in each panel indicates the BLH (RL tops) retrieved from RCS, CNR and vertical wind, respectively.**

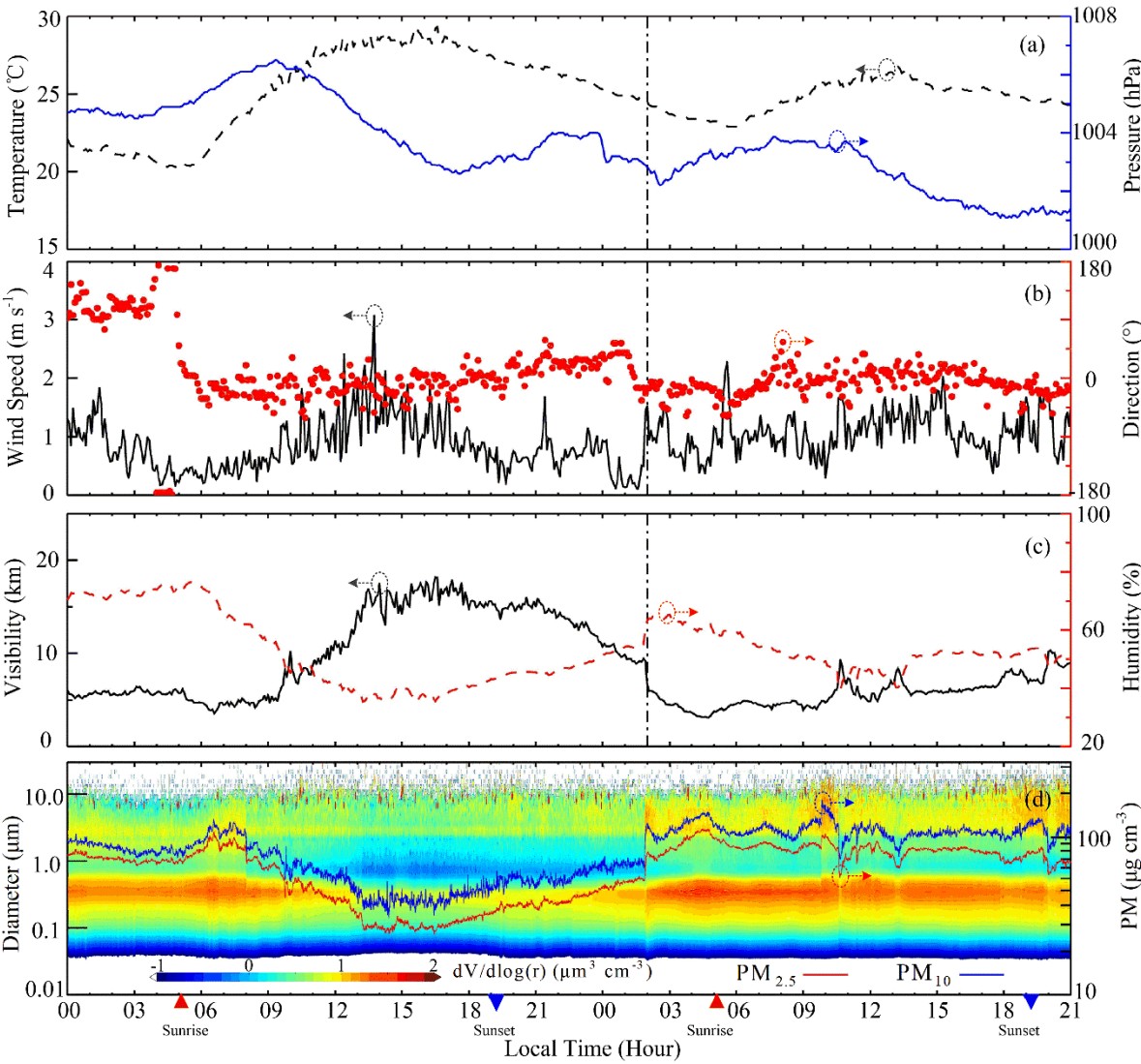

**Figure 4: The simultaneous measured surface meteorological parameters during the experiment from 1 June 2018 to 2 June 2018. From top panel to bottom panel are (a) temperature, pressure; (b) wind velocity, wind direction; (c) visibility, relative humidity; (d) logarithmic aerosol volume size, PM2.5 and PM10 concentration, respectively. The vertical dashed lines in (a)-(c) indicate the time when there is a sudden enhancement of PM$_{2.5}$ in (d).**

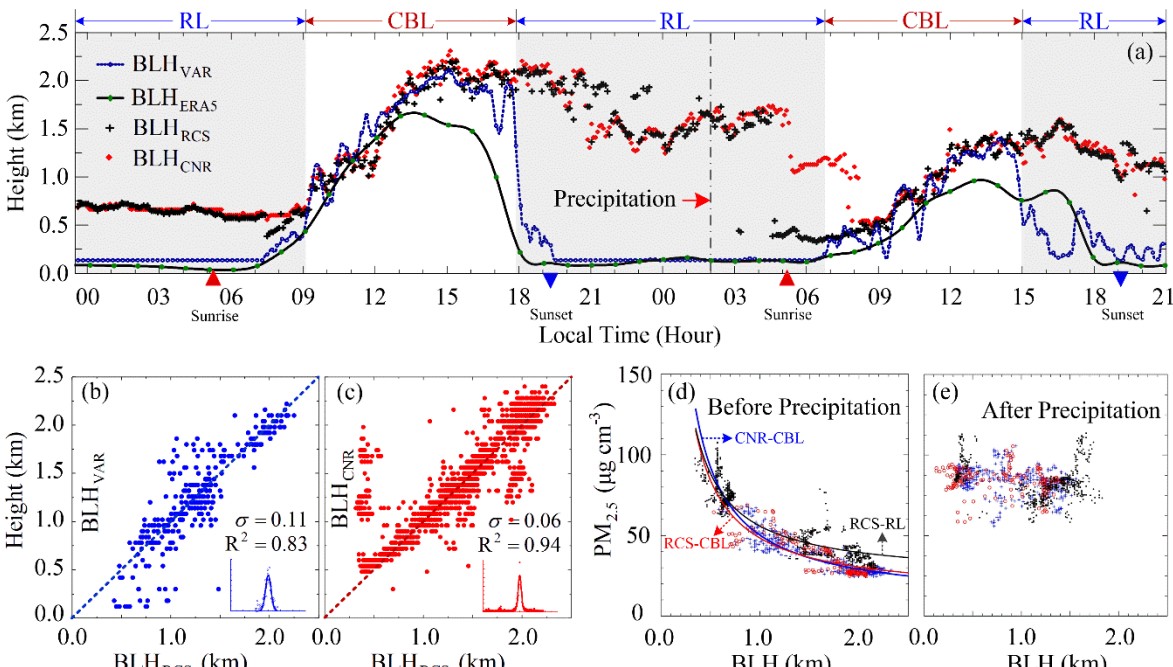

**Figure 5: (a) The BLH retrieval results from different methods. Scatter plots of (b) BLH$_{VAR}$ in CBL and (c) BLH$_{CNR}$ in CBL and RL versus BLH$_{RCS}$ for comparison. The blue and red dashed lines indicate x = y. The BLH differences are plotted in right bottom in (b) and (c). The blue and red solid lines represent the corresponding Gauss fitting. R$^2$ represents the coefficient of determination and σ represents the standard deviation of Gauss fitting. (d) The scatter plot of BLH and PM2.5 before precipitation. Red circles indicate BLH$_{VAR}$ in CBL, while blue pluses indicate BLH$_{RCS}$ in CBL and black dots indicate BLH$_{RCS}$ in RL. The solid lines represent the corresponding inverse fit. (e) Same as (d) but without inverse fitting after precipitation.**

