# Peer review of "Relationship Analysis of PM2.5 and BLH using an Aerosol and Turbulence Detection Lidar"

_Atmospheric Measurement Techniques, 2018_

## Referee Comment (RC1) · Anonymous Referee #1 · 21 Mar 2019

This manuscript presents continuous measurements of PBL structure using a newly-developed compact lidar system combined both direct detection lidar and coherent Doppler win lidar, and demonstrates that the PBL height can be accurately retreated by the measurements and the residual BL and stable BL can be distinguished using different signals from the instrument. The relationships between the PM2.5 concentration and the PBL height were also analyzed. The authors found a strong negative correlation between PM2.5 and PBL height before the precipitation event and a much weaker negative correlation after the precipitation.

The manuscript is well organized and written in general. The instrument is demonstrated as very useful in boundary layer research. The quality of the observations is very impressive. The results and analyses are clear and persuasive. Some of the con-

clusions need to be rephrased under certain contexts. After the following points are addressed, the manuscript is recommended to be published on AMT.

1. Page 1, line 17. Suggest removing "Negative".

2. It is not an ideal location for the weather site to be on the top of a building. The building impacts the temperature, humidity, wind speed and direction. Cautions should be used when analyzing the weather data from such a site.

3. Page 4, line 22. The variance of vertical velocity should just represent the vertical component of the turbulent kinetic energy.

4. Page 5. Model-simulated PBL height in a relatively coarse grid spacing cannot be used to cross-check the observation even though the reanalysis data have assimilated lots of observations. Sounding is probably a better source for observation cross-check. I would recommend the authors to show the 12-hourly sounding data in the city and compare them with the lidar observed PBL structure.

5. Page 6. How strong was the precipitation event? The authors are recommended to provide quantified value of the precipitation either from the met station observation or model-based estimate. The strength of the precipitation impacts the PM2.5 concentration after the event. Usually strong rainfall will scavenge most of the PM2.5 particles while drizzle or light rain can moisten the PBL and facilitate wet growth of smaller aerosols that reach PM2.5.

6. Page 7, line 32. As mentioned in previous comment, there may not be unknown sources but just the wet growth of the existing small particles.

7. Page 14, figure 3. I would recommend the authors to identify RL and SBL tops and if possible, together with the RL bottom at the same time based on the data. These fine structures are extremely useful for model validation and parameterization development.

8. Page 16, figure 5e. These relationships are indeed the result of both cloud effect and precipitation impacts not just precipitation causing the differences. A modeling study is

needed to untangle these two effects and quantify the contributions to the changes of the relationships.

---

## Referee Comment (RC2) · Anonymous Referee #3 · 28 Apr 2019

The manuscript aims to investigate the relationship between BLH and air pollution in different ABL categories. The ABLH is defined based on both a micro-pulse lidar (DDL) and a coherent Doppler wind lidar (CDWL) through wavelet covariance transform method and variance analysis of the vertical velocity. It is well written and the analysis is careful. However, there are some aspects for improvement:

1. Only the relationship between PM2.5 and BLH before and after one precipitation process is analyzed. The manuscript only presents the phenomena, so what accounts for this difference, what role of the precipitation process, it is unclear;

2. ABL may not belong to different categories before and after the precipitation, in fact, according to the Figure 3(a), the growing process of the CBL after the precipitation is very similar to that before the precipitation;

3. From your manuscript, anti-correlation relationship between PM2.5 and BLH is found whether before or after a precipitation. The difference is that the relativity weakened after a precipitation. It seems that precipitation plays an important role. That is, the author paid more attention to different weather conditions instead of "different ABL categories ".

4. The core content of the manuscripts is the "Relationship Analysis of PM2.5 and BLH", from the abstract, only the sentence "Negative correlation between BLH and PM2.5 is analyzed before and after a precipitation." is related to your title. And such conclusion is very common, lower concentration of PM always corresponds to higher BLH if there is no new emission source. The abstract does not show the purpose and innovation point of the study explicitly. Besides, only one paragraph describes the relationship of PM2.5 and BLH in the text? The abstract and the contents of the manuscripts should be improved.

**Some minor revisions are as follows**:

1. For line 3 on page 2, "The boundary layer height (BLH) is the height of the top layer of ABL", the description makes no sense, please improve.

2. For line 9 on page 2: Explain "ABL categories" here.

3. For line 18 on page 2, "Among these instruments, lidar provides sufficient spatial and temporal resolution, long detection range and high accuracy to determine the BLH......", the description should be improved, lidar system provides backscattering signal with sufficient spatial and temporal resolution......

4. For Lines 20-27 on page 2: Here, please highlight the advantages of two lidars.

5. For lines 22-24 on page 2, "in middle atmosphere via Rayleigh scattering......, in mesosphere and lower thermosphere via fluorescence backscatter......." The manuscripts focused on ABL, it may be unnecessary to mention the detection principle in middle atmosphere and in mesosphere and lower thermosphere.

6. For lines 25-26 on page 2, "Recently, a micro-pulse direct detection lidar (DDL) was developed to make continuous measurements of aerosol in troposphere....." In fact, the micro-pulse lidar (MPL) has been widely used to detect ABLH, there are several studies (*He et al., 2008; Sawyer and Li 2013; Li et al., 2017*), not recently, maybe you can describe the advantage of the MPL here, such as detecting with eye-safe laser, small field-of-view removing multiple-layer scattering concerns......
As well as for description about Doppler wind lidar later.

   He Q, Li C, Mao J, et al. Analysis of aerosol vertical distribution and variability in Hong Kong [J]. Journal of Geophysical Research Atmospheres, 2008, 113(D14):-.
   Sawyer, V.; Li, Z.J.A.E.; Detection, variations and intercomparison of the planetary boundary layer depth from radiosonde, lidar and infrared spectrometer. 2013, 79 (11), 518-528.
   Li, H.; Yang, Y.; Hu, X.M.; Huang, Z.; Wang, G.; Zhang, B.J.A.; Application of Convective condensation Level Limiter in Convective Boundary Layer Height Retrieval Based on Lidar Data. 2017, 8 (4), 79

7. For lines 15-16 on page 4, "Considering different vertical spatial resolutions, a dilation of 150 m and 250 m is applied for RCS and CNR, respectively". The selection of an appropriate dilation is the key for WCT method. So why "150 m" and "250 m" are selected? Should be explained.

8. For line 16 on page 4, "Compared with gradient method, HWCT method has greater adjustability and robustness". In fact, as extended technique of gradient method, several studies (*Brooks, 2003; Mao et al., 2013; Dang et al., 2019*) have indicated the WCT method is also easily interference by multiple

aerosol layers or cloud layer. So how the paper ideals with the interference of the cloud layers on ABLH determination in Figure 3(a)-(b)? No doubt, the signal gradient at the cloud boundary is strongest than at the ABL top on 2 June 2018, the HWCT may capture the cloud top rather than the true height of lower stable ABL.

Brooks, I.M.J.J.o.A.; Technology, O.; Finding Boundary Layer Top: Application of Wavelet covariance Transform to Lidar Backscatter Profiles. 2003, 20 (8), 1092—1105.

Mao, F.; Wei, G.; Song, S.; Zhu, Z.; Determination of the boundary layer top from lidar backscatter profiles using a Haar wavelet method over Wuhan, China. Optics Laser Technology 2013, 49 (7), 343-349.

Dang, R.; Yang, Y.; Li, H.; Hu, X.-M.; Wang, Z.; Huang, Z.; Zhou, T.; Zhang, T.; Atmosphere Boundary Layer Height (ABLH) Determination under Multiple-Layer Conditions Using Micro-Pulse Lidar. remote sensing 2019, 11 (263).

9. For line 17 on page 4, "In order to reduce the interference from unexpected turbulence and noise", what is unexpected turbulence? Is the "turbulence" is ambiguous here? Similarly, line 25 on page 4.

10. For lines 19-20 on page 4, "As an example, the measured RCS and CNR after one-minute average (after overlap correction and background noise deduction) at 1 June 2018, 10:40 am is shown in Fig. 2a", Figure 2 shows an example in clear sky situation, profiles in cloudy situations on 2 June 2008 is suggested.

11. For line 22 on page 4, "…….which represented the turbulence kinetic energy", the "represented" should change to "represents".

12. For line 24 on page 4, "In this study, the threshold is set to be 0.06 $m^2s^{-2}$", how the threshold is defined?

13. For line 4 on page 5, "BLH from reanalysis data is always used in boundary layer climatology", please improve the description.

14. For lines 7-8 on page 5, "The hourly BLH from high resolution realisation sub-daily deterministic forecasts of ERA5 is used here", is the ABLH defined from ERA used to estimate the results from lidar? The purpose should be stated. In addition, should "realisation" be changed to "realization"?

15. For line 18 on page 5, "…… indicated the BLH derived from……", "indicated" should be change to "indicate".

16. For lines 17-19 on page 5, the description could be rewritten as "The black dotted line in each panel indicate the BLH derived from RCS, CNR and vertical

wind, called as $BLH_{RCS}$, $BLH_{CNR}$ and $BLH_{VAR}$ in the study".

17. For Line 24 on page 5: From the author, stratocumulus exists above the ABL; It can be seen clearly from Fig. 3(b) that signals between CBL top and cloud are relatively small, and the BLHs derived by aerosol method are cloud heights. Here, the authors should notice the influence of cloud in BLH retrieving based on lidar.

18. For line 28 on page 5, "The results observed in RCS can be also found in CNR", what is the results? The description is unclear.

19. For section 4.1, only the observations of aerosol concentration, the resulted ABLH and meteorological parameters are described, so how do they interact with each other? How does the BLH respond to the meteorological condition?

20. For line 25 on page 6, "in Fig. 3, the BLH results are well retrieved, indicating that the HWCT and variance methods are appropriate for BLH determination……" The HWT and variance analysis may be interfered by the RL and cloud layer, how does this study ideal with the interference of them? Similar to comment 8.

21. For line 29 on page 6, "In turbulence derived CBL, all three BLH results from lidar measurements are comparable when the ABL is fully mixed", please improve the description.

22. For line 31 on page 6, "a criteria is proposed to classify the ABL as CBL and RL/SBL by the values of $BLH_{VAR}$ and $BLH_{RCS}$ in this study……in the morning, when $BLH_{VAR}$ meets the $BLH_{RCS}$, the type of ABL changes from RL/SBL into CBL. In the Afternoon, when $BLH_{VAR}$ departs from $BLH_{RCS}$, the ABL turns into RL/SBL again……." When $BLH_{VAR}$ firstly meets or departs from $BLH_{RCS}$? How to classify if there are several moments that $BLH_{VAR}$ meets or departs from $BLH_{RCS}$?

23. For section 4.3, only the "relationship between the BLH and PM2.5" before and after a precipitation case is analyzed. It is not enough to illustrate the title of the manuscripts. In addition, before precipitation, it is clear that the PM shows a contrary tendency with the ABLH. After precipitation, although the ABLH is lower than on previous day maybe caused by cloud or others, the growing process of CBL is similar to that before precipitation, however, there is no obvious tendency of PM2.5. Therefore, what caused the difference of relationship between PM2.5 and ABLH is the PM2.5 distribution. What should be considered is the factor contributing to the difference of PM before and after precipitation, what's role of the precipitation process?

---

## Author Comment (AC1) · 24 May 2019

We would like to thank the reviewers for their valuable comments and suggestions. We have considered all comments carefully which helped us significantly to improve our manuscript. Following the reviewers' comments and suggestions, we revised the manuscript. Our responses to the reviewers' comments are listed below in blue fonts and the changes in manuscript are listed in *blue italic fonts.*

**Anonymous Referee #1**

This manuscript presents continuous measurements of PBL structure using a newly developed compact lidar system combined both direct detection lidar and coherent Doppler win lidar, and demonstrates that the PBL height can be accurately retreated by the measurements and the residual BL and stable BL can be distinguished using different signals from the instrument. The relationships between the PM2.5 concentration and the PBL height were also analyzed. The authors found a strong negative correlation between PM2.5 and PBL height before the precipitation event and a much weaker negative correlation after the precipitation.

The manuscript is well organized and written in general. The instrument is demonstrated as very useful in boundary layer research. The quality of the observations is very impressive. The results and analyses are clear and persuasive. Some of the conclusions need to be rephrased under certain contexts. After the following points are addressed, the manuscript is recommended to be published on AMT.

Thanks for your positive comments. We have rewritten some conclusions in the revised manuscript.

1. Page 1, line 17. Suggest removing "Negative".

Corrected as suggested.

**Changes:** Page 1, line 18-19. "*Correlation between different BLH and PM$_{2.5}$ is strongly negative before a precipitation event and become much weaker after the precipitation.*"

2. It is not an ideal location for the weather site to be on the top of a building. The building impacts the temperature, humidity, wind speed and direction. Cautions should be used when analyzing the weather data from such a site.

Thanks for this comment. In fact, there is not any ideal location for the weather site in such an urban area. The overly dense buildings will also impact the temperature, humidity, wind speed and direction on the ground. We will pay attention to these cautions when analyzing these data.

**Changes:** Page 6, line 33 - page 7, line 1. "*It should be noted that the building where the instrument deployed would have an impact on these meteorological parameters.*"

3. Page 4, line 22. The variance of vertical velocity should just represent the vertical component of the turbulent kinetic energy.

Corrected as suggested.

**Changes:** Page 5, line 10-11. "*The BLH can also be determined from the variance of vertical velocity $\sigma_w^2$, which represents the vertical component of the turbulence kinetic energy.*"

4. Page 5. Model-simulated PBL height in a relatively coarse grid spacing cannot be used to cross-check the observation even though the reanalysis data have assimilated lots of observations.

Sounding is probably a better source for observation cross-check. I would recommend the authors to show the 12-hourly sounding data in the city and compare them with the lidar observed PBL structure.

Thanks for this suggestion. Unfortunately, there is not any sounding data in Hefei. The nearest sounding station is in Anqing, which is approximately 150 km south to Hefei. A cross-check of BLH retrieved from lidar and sounding data will be carried out in future experiments or observations.

**Changes:** Page 5, line 22-23. "*The hourly BLH from high resolution realisation sub-daily deterministic forecasts of ERA5 is used to cross-check the BLH retrieved from lidar since there is no sounding data in Hefei.*"

5. Page 6. How strong was the precipitation event? The authors are recommended to provide quantified value of the precipitation either from the met station observation or model-based estimate. The strength of the precipitation impacts the PM2.5 concentration after the event. Usually strong rainfall will scavenge most of the PM2.5 particles while drizzle or light rain can moisten the PBL and facilitate wet growth of smaller aerosols that reach PM2.5.

Thanks for your valuable suggestion. There is no precipitation event as drizzle or light rain recorded by the weather transmitter (Vaisala WXT520) or experimenter on the ground or the top of the building. As you suggested, such precipitation event as drizzles above the ground may increase the aerosol, which consists our observation.

**Changes:** Page 7, line 1-2. "*There is no precipitation event recorded on the ground by weather transmitter, even during the precipitation in the cloud as shown in Fig. 3c.*"

6. Page 7, line 32. As mentioned in previous comment, there may not be unknown sources but just the wet growth of the existing small particles.

Thanks for this suggestion. The relationships between BLH and $PM_{2.5}$ are affected after the precipitation. The wet growth of small particles may be responsible during the drizzles. However, during the growing process of CBL after the precipitation, the correlation between BLH and $PM_{2.5}$ is weak even after the drizzles as shown in Fig. 5e and Table 2. This weak relationship when there is no drizzle can't be explained by wet growth of aerosols. Thus, both the pollution sources and meteorological conditions should be considered. We modified this description and added some discussions in the revised manuscript.

**Changes:**

Page 7, line 10-11. "*The wet growth of the existing small particles caused by the precipitation above the ground may be responsible for the sudden increase of aerosols.*"

Page 8, line 23-24. "*The relationships between BLH and PM2.5 are changed after precipitation.*"

7. Page 14, figure 3. I would recommend the authors to identify RL and SBL tops and if possible, together with the RL bottom at the same time based on the data. These fine structures are extremely useful for model validation and parameterization development.

Thanks for this suggestion. We identified RL top as red dotted lines with a temporal resolution of 5 min in revised Fig. 3. The dominant aerosol layer top is retrieved based on threshold method. If the difference between aerosol layer top and BLH is larger than a specified threshold, 0.3 km in current study, the aerosol layer is identified as the RL top. For the SBL top and RL bottom, it is

difficult to be identified due to elevated aerosol layers, e.g., between 1 June 2018 21:00 and 2 June 2018 03:00. For the turbulence derived SBL, the vertical resolution (60 m) of the lidar used in this study is too coarse for an accuracy SBL near the ground (< 200 m). We have developed a new lidar recently, which has higher spatial resolution of meter-scale to solve this problem in future work (Wang et al., 2019). Therefore, we only identified RL tops in the revised manuscript.

**Changes:**

Page 5, line 29-30. "*Compared to the BLH retrieval, RL top can be identified through a simply rough threshold which is described in the Appendix.*"

Page 10, line 5-15. "***Appendix: The RL top retrieval method***

*Besides BLH, RL top is also important in model validation and parameterization development. A simple method to retrieve RL top from RCS, CNR and variance of vertical velocity profiles is proposed. In order to reduce the interference from noise, the RL top is determined with a temporal resolution of 5 min. Dominant aerosol layer tops are easy to be identified around 2 km altitude as shown in Fig, 3. Thus the aerosol layer tops are limited between 1 and 2.5 km altitude range. A threshold method is suitable for RCS and CNR profiles. For this observation, the threshold is set to be $5 \times 10^{10}$ for RCS profile ($1 \times 10^{10}$ for resolution of 1 min as shown in Fig. 3a) and -30 dB for CNR profile. For profiles of variance of vertical velocity, the aerosol layer is identified as the altitudes under the minimum altitude where invalid data exists, e.g., ~1.6 km in Fig. 2c. If the difference between aerosol layer top and BLH is larger than a threshold, e.g., 0.3 km in current study, the aerosol layer top is identified as RL top. It should be noted that all the values of threshold used here may varies at different places for different lidars. These values may be only suitable for during this observation.*"

8. Page 16, figure 5e. These relationships are indeed the result of both cloud effect and precipitation impacts not just precipitation causing the differences. A modeling study is needed to untangle these two effects and quantify the contributions to the changes of the relationships.

Thanks for this comment. This is a good suggestion to probe the different effects by using a model. The reasons for the differences in the relationships may result from both cloud effect and pollutant sources not just precipitations. This study is focus on the relationship between $PM_{2.5}$ and BLH based on a hybrid lidar, not the mechanism of the differences in the relationships. More observational and modeling study are needed to solve this question in future work.

**Changes:**

Page 8, line 23-28. "*The relationships between BLH and PM2.5 are changed after precipitation. Recently, Geißet al. (2017) investigated correlations between BLH and concentrations of pollutants ($PM_{10}$, $O_3$, $NO_x$). They found that the correlations of BLH with $PM_{10}$ were quite different for different sites without showing a clear pattern. In addition, the reflection and absorption of the incoming solar radiation by the clouds on 2 June 2018 could also affect the diffusion of aerosols. Therefore, BLH with different retrieval methods, pollutant sources and meteorological conditions should be considered in air quality prediction models.*"

Page 9, line 23-24. "*The reasons for the differences in the relationships between BLH and $PM_{2.5}$ may result from both cloud effect and pollutant sources not just the precipitation.*"

Page 9, line 26-27. "*To probe the mechanism of the BLH-$PM_{2.5}$ relations under different conditions, such as before and after the precipitation, not only such observations, but also model simulation are needed in further studies.*"

**References:**

Wang, C., Xia, H., Wu, Y., Dong, J., Wei, T., Wang, L., and Dou, X.: Meter-scale spatial-resolution-coherent Doppler wind lidar based on Golay coding, Optics letters, 44, 311-314, 10.1364/OL.44.000311, 2019.

---

## Author Comment (AC2) · 24 May 2019

We would like to thank the reviewers for their valuable comments and suggestions. We have considered all comments carefully which helped us significantly to improve our manuscript. Following the reviewers' comments and suggestions, we revised the manuscript. Our responses to the reviewers' comments are listed below in blue fonts and the changes in manuscript are listed in *blue italic fonts*.

**Anonymous Referee #3**

The manuscript aims to investigate the relationship between BLH and air pollution in different ABL categories. The ABLH is defined based on both a micro-pulse lidar (DDL) and a coherent Doppler wind lidar (CDWL) through wavelet covariance transform method and variance analysis of the vertical velocity. It is well written and the analysis is careful. However, there are some aspects for improvement:

Thanks for your careful and thoughtful comments. We revised the manuscript according to your suggestions.

1. Only the relationship between PM2.5 and BLH before and after one precipitation process is analyzed. The manuscript only presents the phenomena, so what accounts for this difference, what role of the precipitation process, it is unclear;

The precipitation event above the ground may be responsible for the sudden increase of aerosol due to wet growth of smaller aerosols. The precipitation may lead to this difference in the early hours after the precipitation. *Geiß et al.*, 2017 investigated the relashionship between BLH and $PM_{10}$. They found that the pollution sources, meteorological conditions and BLH retrieval details should be considered. In addition, the cloud effect should also be considered. Thus a complex process which is unknown accounts for this difference. More observations in under different conditions and modeling study would be helpful to improve our knowledge on this complex topic.

**Changes:**

Page 8, line 23-28. "*The relationships between BLH and PM2.5 are changed after precipitation. Recently, Geiß et al. (2017) investigated correlations between BLH and concentrations of pollutants ($PM_{10}$, $O_3$, $NO_x$). They found that the correlations of BLH with $PM_{10}$ were quite different for different sites without showing a clear pattern. In addition, the reflection and absorption of the incoming solar radiation by the clouds on 2 June 2018 could also affect the diffusion of aerosols. Therefore, BLH with different retrieval methods, pollutant sources and meteorological conditions should be considered in air quality prediction models.*"

Page 9, line 23-24. "*The reasons for the differences in the relationships between BLH and $PM_{2.5}$ may result from both cloud effect and pollutant sources not just the precipitation.*"

Page 9, line 26-27. "*To probe the mechanism of the BLH-$PM_{2.5}$ relations under different conditions, such as before and after the precipitation, not only such observations, but also model simulation are needed in further studies.*"

2. ABL may not belong to different categories before and after the precipitation, in fact, according to the Figure 3(a), the growing process of the CBL after the precipitation is very similar to that before the precipitation;

Yes, the growing processes of the CBL before and after the precipitation are similar. In Sect.

4.3, we mentioned that "*In general, these results show good responses of $PM_{2.5}$ to aerosol derived BLH ($BLH_{RCS}$) evolution with larger $R^2$ and stronger correlation than turbulence derived BLH ($BLH_{VAR}$) both before and after precipitation.*" The different ABL categories in this manuscript mean that aerosol derived BLH (static, i.e., $BLH_{RCS}$ and $BLH_{CNR}$) and turbulence derived BLH (dynamical, i.e., $BLH_{VAR}$). We are very sorry for this confusing expression and modified it in the revised manuscript.

**Changes:** Page 8, line 16-18. "*In general, these results show good responses of $PM_{2.5}$ to aerosol derived BLH ($BLH_{RCS}$) evolution with larger $R^2$ and stronger correlation than turbulence derived BLH ($BLH_{VAR}$) both before and after precipitation.*"

3. From your manuscript, anti-correlation relationship between PM2.5 and BLH is found whether before or after a precipitation. The difference is that the relativity weakened after a precipitation. It seems that precipitation plays an important role. That is, the author paid more attention to different weather conditions instead of "different ABL categories ".

Thanks for this comment. As answered to comment 2, the different ABL categories are ABL retrieved from aerosol signal and turbulence, respectively. We have discussed the different relationships between $PM_{2.5}$ and BLH under different ABL categories ($BLH_{RCS}$ and $BLH_{VAR}$) in Sect.4.3 and Table 2. We apologize for this confusing expression in the manuscript.

**Changes:**

Page 2, line 16-18. "*However, the relationship analysis of $PM_{2.5}$ and BLH in different ABL categories, i.e., aerosol derived (static) BLH and turbulence derived (dynamical) BLH, is still rare.*"

Page 8, line 16-18. "*In general, these results show good responses of $PM_{2.5}$ to aerosol derived BLH ($BLH_{RCS}$) evolution with larger $R^2$ and stronger correlation than turbulence derived BLH ($BLH_{VAR}$) both before and after precipitation.*"

4. The core content of the manuscripts is the "Relationship Analysis of PM2.5 and BLH", from the abstract, only the sentence "Negative correlation between BLH and PM2.5 is analyzed before and after a precipitation." is related to your title. And such conclusion is very common, lower concentration of PM always corresponds to higher BLH if there is no new emission source. The abstract does not show the purpose and innovation point of the study explicitly. Besides, only one paragraph describes the relationship of PM2.5 and BLH in the text? The abstract and the contents of the manuscripts should be improved.

Thanks for your comment. We revised the title as "Relationship Analysis of $PM_{2.5}$ and BLH using an Aerosol and Turbulence Detection Lidar". The relationship analysis is based on this innovative hybrid lidar. The advantages of this hybrid lidar is introduced in responses to minor comment 4 and 6. Then, the BLH retrieval method and retrieved BLH results should be evaluated. Finally, the relationship analysis can be performed. Thus, all of these are related to the title, not only one paragraph. In previous work, the correlation could be negative, but also positive (*Geiß et al.*, 2017). So comparing the correlation under different conditions and places in the world is desired to improve our understanding of this complex topic: relashionship between PM and BLH. The relashionship analysis before and after precipitation in this study may be helpful to this complex topic. We also revised the abstract and the contents according to your suggestions.

**Changes:**

Page 1, line 1-2. "*Relationship Analysis of $PM_{2.5}$ and BLH using an Aerosol and Turbulence*

*Detection Lidar"*

Page 1, line 18-20. "*Correlation between different BLH and PM$_{2.5}$ is strongly negative before a precipitation event and become much weaker after the precipitation. Different relations between PM$_{2.5}$ and BLH may result from different BLH retrieval methods, pollutant sources and meteorological conditions.*"

Page 8, line 23-28. "*The relationships between BLH and PM2.5 are changed after precipitation. Recently, Geiß et al. (2017) investigated correlations between BLH and concentrations of pollutants (PM$_{10}$, O$_3$, NO$_x$). They found that the correlations of BLH with PM$_{10}$ were quite different for different sites without showing a clear pattern. In addition, the reflection and absorption of the incoming solar radiation by the clouds on 2 June 2018 could also affect the diffusion of aerosols. Therefore, BLH with different retrieval methods, pollutant sources and meteorological conditions should be considered in air quality prediction models.*"

Page 9, line 23-24. "*The reasons for the differences in the relationships between BLH and PM$_{2.5}$ may result from both cloud effect and pollutant sources not just the precipitation.*"
Page 9, line 26-27. "*To probe the mechanism of the BLH-PM$_{2.5}$ relations under different conditions, such as before and after the precipitation, not only such observations, but also model simulation are needed in further studies.*"

Some minor revisions are as follows:
1. For line 3 on page 2, "The boundary layer height (BLH) is the height of the top layer of ABL", the description makes no sense, please improve.

Deleted.

2. For line 9 on page 2: Explain "ABL categories" here.

Thanks for this comment. The ABL categories are explained in the revised manuscript. "*different ABL categories, i.e., aerosol derived (static) BLH and turbulence derived (dynamical) BLH*"

**Changes:** Page 2, line 16-18. "*However, the relationship analysis of PM$_{2.5}$ and BLH in different ABL categories, i.e., aerosol derived (static) BLH and turbulence derived (dynamical) BLH, is still rare.*"

3. For line 18 on page 2, "Among these instruments, lidar provides sufficient spatial and temporal resolution, long detection range and high accuracy to determine the BLH……", the description should be improved, lidar system provides backscattering signal with sufficient spatial and temporal resolution……

Corrected as suggested.
**Changes:** Page 2, line 16-18. "*Among these instruments, lidar system provides backscattering signal with sufficient spatial and temporal resolution, long detection range and high accuracy to determine the BLH.*"

4. For Lines 20-27 on page 2: Here, please highlight the advantages of two lidars.

Thanks for this suggestion. We added some advantages of these two lidars.
**Changes:**
Page 2, line 25-26. "*Recently, a micro-pulse direct detection lidar (DDL) based on up-*

*conversion technology was developed to make continuous measurements of aerosol in troposphere (Xia et al., 2015)*"

Page 2, line 27-32. "*Different from traditional micro-pulse lidars operated at or near 532 nm (He et al., 2008; Li et al., 2017b; Sawyer and Li, 2013), these two lidars are operated at 1.5 μm, which are eye-safe and can be made with all-fiber components. The 1.5 μm laser shows the highest maximum permissible exposure in the wavelength range from 0.3 to 10 μm (Xia et al., 2015). The invisible infrared eye-safe laser makes these two lidars can work in a densely populated city horizontally. The all-fiber structure makes these lidars robust, immune to external environment changes such as vibration and temperature.*"

5. For lines 22-24 on page 2, "in middle atmosphere via Rayleigh scattering……, in mesosphere and lower thermosphere via fluorescence backscatter……." The manuscripts focused on ABL, it may be unnecessary to mention the detection principle in middle atmosphere and in mesosphere and lower thermosphere.

Deleted as suggested.

6. For lines 25-26 on page 2, "Recently, a micro-pulse direct detection lidar (DDL) was developed to make continuous measurements of aerosol in troposphere….." In fact, the micro-pulse lidar (MPL) has been widely used to detect ABLH, there are several studies (He et al., 2008; Sawyer and Li 2013; Li et al., 2017), not recently, maybe you can describe the advantage of the MPL here, such as detecting with eye-safe laser, small field-of-view removing multiple-layer scattering concerns…… As well as for description about Doppler wind lidar later.

He Q, Li C, Mao J, et al. Analysis of aerosol vertical distribution and variability in Hong Kong [J]. Journal of Geophysical Research Atmospheres, 2008, 113(D14):-.

Sawyer, V.; Li, Z.J.A.E.; Detection, variations and intercomparison of the planetary boundary layer depth from radiosonde, lidar and infrared spectrometer. 2013, 79 (11), 518-528.

Li, H.; Yang, Y.; Hu, X.M.; Huang, Z.; Wang, G.; Zhang, B.J.A.; Application of Convective condensation Level Limiter in Convective Boundary Layer Height Retrieval Based on Lidar Data. 2017, 8 (4), 79

Thanks for this suggestion. The micro-pulse direct detection lidar (DDL) is "*based on up-conversion technology*". "*Different from traditional micro-pulse lidars operated at or near 532 nm (He et al., 2008; Li et al., 2017b; Sawyer and Li, 2013), these two lidars are operated at 1.5 μm, which are eye-safe and can be made with all-fiber components. The 1.5 μm laser shows the highest maximum permissible exposure in the wavelength range from 0.3 to 10 μm (Xia et al., 2015). The invisible infrared eye-safe laser makes these two lidars can work in a densely populated city horizontally. The all-fiber structure makes these lidars robust, immune to external environment changes such as vibration and temperature*." Then the two lidars are integrated into one lidar system. "*In this work, a hybrid lidar integrating both systems are developed for simultaneous measurements of aerosol and vertical wind.*" The advantages of this hybrid lidar has been described in Sect. 2.1. "*Two lidar systems use only one set of laser source, optical collimator and control system. The unique optical telescope guarantees that the measured signal in both systems are from the same backscattering volume, and the radial wind profile and aerosol concentration are measured simultaneously.*" We also showed the advantages of this hybrid lidar in the abstract and conclusions in the revised manuscript.

**Changes:**

Page 1, line 12-13. "*This hybrid lidar is operated at 1.5 μm which is eye-safe and is made of all-fiber components.*"

Page 2, line 25-26. "*Recently, a micro-pulse direct detection lidar (DDL) based on up-conversion technology was developed to make continuous measurements of aerosol in troposphere (Xia et al., 2015)*"

Page 2, line 27-32. "*Different from traditional micro-pulse lidars operated at or near 532 nm (He et al., 2008; Li et al., 2017b; Sawyer and Li, 2013), these two lidars are operated at 1.5 μm, which are eye-safe and can be made with all-fiber components. The 1.5 μm laser shows the highest maximum permissible exposure in the wavelength range from 0.3 to 10 μm (Xia et al., 2015). The invisible infrared eye-safe laser makes these two lidars can work in a densely populated city horizontally. The all-fiber structure makes these lidars robust, immune to external environment changes such as vibration and temperature.*"

Page 9, line 8-11. "*The DDL incorporated a fiber laser at 1.5 μm and an up-conversion detector. This design of lidar makes it more eye-safe than traditional laser of 355, 532 and 1064 nm. All-fiber configuration is realized to guarantee the high optical coupling efficiency and robust stability. Two lidar systems use only one set of laser source, optical collimator and control system.*"

[Figure]

**Figure R1**. WCT method with different values of dilation for RCS. The values of dilation are 100 m, 150 m and 200 m for upper panels, 300 m, 400 m and 500 m for bottom panels, from left to right. The colored contours indicate the WCT results. The black dotted lines indicate retrieved BLH.

7. For lines 15-16 on page 4, "Considering different vertical spatial resolutions, a dilation of 150 m and 250 m is applied for RCS and CNR, respectively". The selection of an appropriate dilation is the key for WCT method. So why "150 m" and "250 m" are selected? Should be explained.

Thanks for this suggestion. We fully agree with your point of view "The selection of an appropriate dilation is the key for WCT method". Too large or too small dilation is not appropriate. We have tested different values of dilation as shown in Fig. R1, even height-dependent dilation that selected by previous studies for WCT method. At least for this 45 hour observations from 1 June to 2 June in Hefei, 150 m is one of the most appropriate values of dilation for RCS. The 250 m for CNR is similar. In fact, the optimum value is equal to the depth of the transition zone (Brooks, 2003). The depth of transition zone varies in different places and seasons. A further study of transition zone depth is desirable by multi instruments with longer enough observations.

**Changes:** Page 4, line 21-22. "*Considering different vertical spatial resolutions and having*

*tested multi values of dilation, a dilation of 150 m and 250 m is applied for RCS and CNR, respectively for this 45-hour observations.*"

8. For line 16 on page 4, "Compared with gradient method, HWCT method has greater adjustability and robustness". In fact, as extended technique of gradient method, several studies (Brooks, 2003; Mao et al., 2013; Dang et al., 2019) have indicated the WCT method is also easily interference by multiple aerosol layers or cloud layer. So how the paper ideals with the interference of the cloud layers on ABLH determination in Figure 3(a)-(b)? No doubt, the signal gradient at the cloud boundary is strongest than at the ABL top on 2 June 2018, the HWCT may capture the cloud top rather than the true height of lower stable ABL.

Brooks, I.M.J.J.o.A.; Technology, O.; Finding Boundary Layer Top: Application of Wavelet covariance Transform to Lidar Backscatter Profiles. 2003, 20 (8), 1092—1105.

Mao, F.; Wei, G.; Song, S.; Zhu, Z.; Determination of the boundary layer top from lidar backscatter profiles using a Haar wavelet method over Wuhan, China. Optics Laser Technology 2013, 49 (7), 343-349.

Dang, R.; Yang, Y.; Li, H.; Hu, X.-M.; Wang, Z.; Huang, Z.; Zhou, T.; Zhang, T.; Atmosphere Boundary Layer Height (ABLH) Determination under Multiple-Layer Conditions Using Micro-Pulse Lidar. remote sensing 2019, 11 (263).

Thanks for this comment. For the cloud layer and aerosol layer higher than 2.5 km as shown in Fig. 3, we can easily remove the interference of such cloud layers above ABL by setting a top-limit of the WCT method in this manuscript similar to Dang et al., 2019. For the multiple aerosol layers in the ABL, an appropriate dilation is useful and robust as shown in Fig. R1. For the scattered stratocumulus that exist in the capping layer as shown in Fig. 3a and 3b, the difference between cloud top and BLH are relatively small. In addition, the duration time of stratocumulus is also short in the field of view of the lidar that can be easily removed by a longer temporal resolution. Thus the influence of scattered stratocumulus is negligible. For the continuous thick low level cloud not shown in this observation, the BLH cannot be retrieved. Thus, the interference of the cloud layers and multiple aerosol layers are negligible at least in this manuscript. We added some description of the cloud in Sect. 3 in the revised manuscript.

**Changes:** Page 5, line 1-6. "*It should be noted that cloud layer could affect the BLH results. A top-limit is set to the HWCT method for higher clouds. For the scattered stratocumulus that may exist in the capping layer, the differences between cloud top and BLH are relatively small. In addition, the duration time of stratocumulus is also short in the field of view of the lidar. Thus the influence of scattered stratocumulus is negligible. The low level cloud in the ABL can be identified by the paired minimum $W_f(a,b)$ and maximum $W_f(a,b)$ occurs at heights close to each other. The BLH cannot be retrieved under this condition.*"

9. For line 17 on page 4, "In order to reduce the interference from unexpected turbulence and noise", what is unexpected turbulence? Is the "turbulence" is ambiguous here? Similarly, line 25 on page 4.

Thanks for this comment. We removed "turbulence" here in line 17. But in line 25, the unexpected turbulence means turbulence occurs in the free atmosphere where no turbulence is considered to exists.

**Changes:**

Page 4, line 24 – page 5, line 1. "*In order to reduce the interference from unexpected noise, the signal is averaged to a temporal resolution of 1 min in BLH determination.*"

Page 5, line 13-14. "*A median algorithm is used to mitigate the interference and fluctuation from unexpected turbulence and noise in the free atmosphere*"

10. For lines 19-20 on page 4, "As an example, the measured RCS and CNR after one-minute average (after overlap correction and background noise deduction) at 1 June 2018, 10:40 am is shown in Fig. 2a", Figure 2 shows an example in clear sky situation, profiles in cloudy situations on 2 June 2008 is suggested.

Thanks for this suggestion. As answered to minor comment 8, the interference of the clouds is removed by setting a top-limit of 2.5 km in this manuscript. Besides, to propose a robust BLH retrieval method under complex conditions is beyond the scope of current manuscript, but such work is desirable with more observations in future.

**Changes:**

Page 5, line 2. "*A top-limit is set to the HWCT method for higher clouds.*"

Page 7, line 15. "*A top-limit of 2.5 km of BLH is applied during the BLH retrieval.*"

11. For line 22 on page 4, "…which represented the turbulence kinetic energy", the "represented" should change to "represents".

Corrected as suggested.

**Changes:** Page 5, line 10-11. "*The BLH can also be determined from the variance of vertical velocity $\sigma_w^2$, which represents the vertical component of the turbulence kinetic energy.*"

12. For line 24 on page 4, "In this study, the threshold is set to be 0.06 m2s-2", how the threshold is defined?

Similar to that of dilation, we have tested different values of threshold for this observation. The variance of vertical velocity with 5 min temporal resolution is shown in Fig. R2. A threshold between 0.04 $m^2s^{-2}$ and 0.15 $m^2s^{-2}$ may be appropriate. As shown in Fig. 2c, 0.06 $m^2s^{-2}$ is one of the most appropriate threshold during this observation. A smaller value may be difficult to identify free atmosphere while a larger value may be difficult to distinguish CBL with several lower variances,

[Figure]

**Figure R2**. Variance of vertical velocity with 5 min temporal resolution.

such as the profiles shown in Fig. 2c. It should also be noted that the threshold may varies with different places and seasons.

**Changes:** Page 5, line 12-13. "*In this study, the threshold is set to be 0.06 $m^2s^{-2}$ which is suitable as shown in Fig. 2c.*"

13. For line 4 on page 5, "BLH from reanalysis data is always used in boundary layer climatology", please improve the description.

We modified this sentence as follows: "*Reanalysis data is always used in climatological and regional analysis of BLH (Collaud Coen et al., 2014; Guo et al., 2016; Seidel et al., 2012).*"

**Changes:** Page 5, line 18-19. "*Reanalysis data is always used in climatological and regional analysis of BLH (Collaud Coen et al., 2014; Guo et al., 2016; Seidel et al., 2012).*"

14. For lines 7-8 on page 5, "The hourly BLH from high resolution realization sub-daily deterministic forecasts of ERA5 is used here", is the ABLH defined from ERA used to estimate the results from lidar? The purpose should be stated. In addition, should "realisation" be changed to "realization"?

Yes, the hourly BLH from high resolution realisation sub-daily deterministic forecasts of ERA5 is used to cross-check the BLH retrieved from lidar since there is no sounding data in Hefei. The use of "high resolution realization" can be seen from ECMWF website at https://confluence.ecmwf.int/display/CKB/ERA5+data+documentation, the first sentence of third paragraph in the Introduction, "The ERA5 dataset contains one (31 km) high resolution realisation (HRES) and a reduced resolution ten member ensemble (EDA)".

**Changes:** Page 5, line 22-23. "*The hourly BLH from high resolution realisation sub-daily deterministic forecasts of ERA5 is used to cross-check the BLH retrieved from lidar since there is no sounding data in Hefei.*"

15. For line 18 on page 5, "...... indicated the BLH derived from......", "indicated" should be change to "indicate".

Corrected as suggested.

**Changes:** Page 6, line 4-5. "*The black dotted line in each panel indicate the BLH derived from RCS, CNR and vertical wind, called as $BLH_{RCS}$, $BLH_{CNR}$ and $BLH_{VAR}$ in this study.*"

16. For lines 17-19 on page 5, the description could be rewritten as "The black dotted line in each panel indicate the BLH derived from RCS, CNR and vertical wind, called as BLHRCS, BLHCNR and BLHVAR in the study".

Corrected as suggested.

**Changes:** Page 6, line 4-5. "*The black dotted line in each panel indicate the BLH derived from RCS, CNR and vertical wind, called as $BLH_{RCS}$, $BLH_{CNR}$ and $BLH_{VAR}$ in this study.*"

17. For Line 24 on page 5: From the author, stratocumulus exists above the ABL; It can be seen clearly from Fig. 3(b) that signals between CBL top and cloud are relatively small, and the BLHs derived by aerosol method are cloud heights. Here, the authors should notice the influence of cloud in BLH retrieving based on lidar.

Thanks for this suggestion. As answered to minor comment 8, the influence of scattered

stratocumulus exist in the capping layer with short duration time is negligible in this manuscript. The low level cloud is not exits during this observation. The low level cloud in the ABL can be identified by the paired minimum $W_f(a, b)$ and maximum $W_f(a, b)$ occurs at heights close to each other. The BLH cannot be retrieved under this condition.

**Changes:** Page 5, line 1-6. "*It should be noted that cloud layer could affect the BLH results. A top-limit is set to the HWCT method for higher clouds. For the scattered stratocumulus that may exist in the capping layer, the differences between cloud top and BLH are relatively small. In addition, the duration time of stratocumulus is also short in the field of view of the lidar. Thus the influence of scattered stratocumulus is negligible. The low level cloud in the ABL can be identified by the paired minimum $W_f(a, b)$ and maximum $W_f(a, b)$ occurs at heights close to each other. The BLH cannot be retrieved under this condition.*"

18. For line 28 on page 5, "The results observed in RCS can be also found in CNR", what is the results? The description is unclear.

We modified this description in revised manuscript. "*The phenomena that observed in RCS described above can be also found in CNR.*"

**Changes:** Page 6, line 16-17. "*The phenomena that observed in RCS described above can be also found in CNR.*"

19. For section 4.1, only the observations of aerosol concentration, the resulted ABLH and meteorological parameters are described, so how do they interact with each other? How does the BLH respond to the meteorological condition?

This is a good point. However, it's a complex question that beyond the scope of this manuscript. This paper focuses on the relationship between $PM_{2.5}$ and BLH based on a hybrid lidar in this manuscript. In fact, the meteorological parameters described here are intended to explain the evolution of PM, not the interaction with ABL. These parameters are essential for the study of correlations between BLH and PM concentrations (*Geiß et al.*, 2017). Nevertheless, it seems that there is a strong positive correlation between BLH and temperature. The maximum BLH is lower on 2 June 2018 than that on 1 June 2018, so does the maximum temperature. But this correlation may be due to the cloud-ABL interaction as discussed in Sect. 4.4. A recently accepted work on GRL may be helpful to this question (*Guo et al.*, 2019). The influence of meteorology on the BLH has been investigated using long-term (1979-2016) radiosonde data in this work. More observational and modeling study are needed in future work.

20. For line 25 on page 6, "in Fig. 3, the BLH results are well retrieved, indicating that the HWCT and variance methods are appropriate for BLH determination……" The HWT and variance analysis may be interfered by the RL and cloud layer, how does this study ideal with the interference of them? Similar to comment 8.

For the HWCT method, we have answered how to deal with cloud layers in response to minor comment 8. An appropriate value of dilation of HWCT may be enough to deal with interference of RL as shown in Fig. R1, and so does an appropriate threshold of variance method as shown in Fig. R2. For the variance method, there is not any interference of cloud layer above the ABL. The low-level cloud that may interfere the results of variance method is not occurred during this observation.

**Changes:** Page 5, line 1-6. "*It should be noted that cloud layer could affect the BLH results. A*

*top-limit is set to the HWCT method for higher clouds. For the scattered stratocumulus that may exist in the capping layer, the differences between cloud top and BLH are relatively small. In addition, the duration time of stratocumulus is also short in the field of view of the lidar. Thus the influence of scattered stratocumulus is negligible. The low level cloud in the ABL can be identified by the paired minimum $W_f(a, b)$ and maximum $W_f(a, b)$ occurs at heights close to each other. The BLH cannot be retrieved under this condition.*"

21. For line 29 on page 6, "In turbulence derived CBL, all three BLH results from lidar measurements are comparable when the ABL is fully mixed", please improve the description.

We modified this sentence as follows: "*All three retrieved BLH from lidar measurements are comparable when the ABL is fully mixed.*"

**Changes:** Page 7, line 18-19. "*All three retrieved BLH from lidar measurements are comparable when the ABL is fully mixed.*"

22. For line 31 on page 6, "a criteria is proposed to classify the ABL as CBL and RL/SBL by the values of BLHVAR and BLHRCS in this study……in the morning, when BLHVAR meets the BLHRCS, the type of ABL changes from RL/SBL into CBL. In the Afternoon, when BLHVAR departs from BLHRCS, the ABL turns into RL/SBL again……." When BLHVAR firstly meets or departs from BLHRCS? How to classify if there are several moments that BLHVAR meets or departs from BLHRCS?

Here we defined $\Delta$= $BLH_{RCS}$ - $BLH_{VAR}$. There is a hypothesis that $BLH_{RCS}$ (RL) is higher than $BLH_{VAR}$ (SBL) at midnight. This hypothesis is true in most cases. In the morning, the $BLH_{VAR}$ grows with temperature increases. The meet (depart) is defined as when the sign of $\Delta$ become negative (positive) for the first (last) time after (before) midnight. If the sign of $\Delta$ never changes during the whole day and night, a specified value is used. The meet (depart) is defined as when the value of $\Delta$ is less than (greater than) the specified value for the first (last) time after (before) midnight. It should be noted that the specified value varies with different places and seasons. We added more description in the revised manuscript.

**Changes:** Page 7, line 21-26. "*A parameter is defined as $\Delta$= $BLH_{RCS} - BLH_{VAR}$. The sign of $\Delta$ is positive at nighttime in most cases. In the evening, a SBL is capped by a RL as shown in Fig. 5a. In the morning, when $BLH_{VAR}$ meets the value of $BLH_{RCS}$, i.e., the sign of $\Delta$ become negative or the value of $\Delta$ is less than a specified value for the first time after midnight, the type of ABL changes from RL/SBL into CBL. In the Afternoon, when $BLH_{VAR}$ departs from $BLH_{RCS}$, i.e., the sign of $\Delta$ become positive or the value of $\Delta$ is greater than a specified value for the last time before midnight, the ABL turns into RL/SBL again.*"

23. For section 4.3, only the "relationship between the BLH and PM2.5" before and after a precipitation case is analyzed. It is not enough to illustrate the title of the manuscripts. In addition, before precipitation, it is clear that the PM shows a contrary tendency with the ABLH. After precipitation, although the ABLH is lower than on previous day maybe caused by cloud or others, the growing process of CBL is similar to that before precipitation, however, there is no obvious tendency of PM2.5. Therefore, what caused the difference of relationship between PM2.5 and ABLH is the PM2.5 distribution. What should be considered is the factor contributing to the difference of PM before and after precipitation, what's role of the precipitation process?

Besides the precipitation, the relationship between BLH and $PM_{2.5}$ is also analyzed with

different BLH retrieval methods as shown in Fig. 5d~5e and Table2. A good response of PM$_{2.5}$ to aerosol derived BLH evolution with larger R$^2$ and stronger correlation than turbulence derived BLH both before and after the precipitation. The correlation between BLH and PM$_{2.5}$ becomes much weaker after the precipitation. The wet growth of existing small particles caused by the precipitation process may be responsible in the early hours. Recently, Geiß et al. (2017) investigated correlations between BLH and concentrations of pollutants (PM$_{10}$, O$_3$, NO$_x$). They found that the correlations of BLH with PM10 were quite different for different sites without showing a clear pattern. The pollution sources, meteorological conditions and details of BLH retrievals should be considered (*Geiß et al.*, 2017). More observational and modeling study are need to solve this question in future work. We added some discussions in the revised manuscript.

**Changes:**

Page 8, line 23-28. "*The relationships between BLH and PM2.5 are changed after precipitation. Recently, Geiß et al. (2017) investigated correlations between BLH and concentrations of pollutants (PM$_{10}$, O$_3$, NO$_x$). They found that the correlations of BLH with PM$_{10}$ were quite different for different sites without showing a clear pattern. In addition, the reflection and absorption of the incoming solar radiation by the clouds on 2 June 2018 could also affect the diffusion of aerosols. Therefore, BLH with different retrieval methods, pollutant sources and meteorological conditions should be considered in air quality prediction models.*"

Page 9, line 23-24. "*The reasons for the differences in the relationships between BLH and PM$_{2.5}$ may result from both cloud effect and pollutant sources not just the precipitation.*"

Page 9, line 26-27. "*To probe the mechanism of the BLH-PM$_{2.5}$ relations under different conditions, such as before and after the precipitation, not only such observations, but also model simulation are needed in further studies.*"

**References:**

Brooks, I. M., Finding Boundary Layer Top: Application of a Wavelet Covariance Transform to Lidar Backscatter Profiles, Journal of Atmospheric and Oceanic Technology, 20(8), 1092-1105, doi:10.1175/1520-0426(2003)020<1092:fbltao>2.0.co;2, 2003.

Geiß, A., M. Wiegner, B. Bonn, K. Schäfer, R. Forkel, E. von Schneidemesser, C. Münkel, K. L. Chan, and R. Nothard, Mixing layer height as an indicator for urban air quality?, Atmospheric Measurement Techniques, 10(8), 2969-2988, doi:10.5194/amt-10-2969-2017, 2017.

Guo, J., Li, Y., Cohen, J. B., Li, J., Chen, D., Xu, H., Liu, L., Yin, J., Hu, K., Zha, P.: Shift in the temporal trend in boundary layer height trend in China using long-term (1979–2016) radiosonde data, Geophysical Research Letters, doi:10.1029/2019GL082666